# Sepsis Encephalopathy Is Partly Mediated by miR370-3p-Induced Mitochondrial Injury but Attenuated by BAM15 in Cecal Ligation and Puncture Sepsis Male Mice

**DOI:** 10.3390/ijms23105445

**Published:** 2022-05-13

**Authors:** Pratsanee Hiengrach, Peerapat Visitchanakun, Pakteema Tongchairawewat, Ponphisudti Tangsirisatian, Thitiphat Jungteerapanich, Patcharee Ritprajak, Dhammika Leshan Wannigama, Pattarin Tangtanatakul, Asada Leelahavanichkul

**Affiliations:** 1Center of Excellence on Translational Research in Inflammation and Immunology (CETRII), Department of Microbiology, Chulalongkorn University, Bangkok 10330, Thailand; pratsaneeh@gmail.com (P.H.); peerapat.visitchanakun@gmail.com (P.V.); 2Chulalongkorn University International Medical Program, Faculty of Medicine, Chulalongkorn University, Bangkok 10330, Thailand; ptong@docchula.com (P.T.); p.tangsirisatian@docchula.com (P.T.); mook@docchula.com (T.J.); 3Research Unit in Integrative Immuno-Microbial Biochemistry and Bioresponsive Nanomaterials, Department of Microbiology, Faculty of Dentistry, Chulalongkorn University, Bangkok 10330, Thailand; patcharee.r@chula.ac.th; 4Antimicrobial Resistance and Stewardship Research Unit, Department of Microbiology, Faculty of Medicine, Chulalongkorn University, Bangkok 10330, Thailand; dhammika.l@chula.ac.th; 5School of Medicine, Faculty of Health and Medical Sciences, The University of Western Australia, Nedlands, WA 6009, Australia; 6Department of Transfusion Medicine and Clinical Microbiology, Faculty of Allied Health Sciences, Chulalongkorn University, Bangkok 10330, Thailand; 7Center of Excellence in Immunology and Immune-Mediated Disease, Department of Microbiology, Chulalongkorn University, Bangkok 10330, Thailand; 8Nephrology Unit, Department of Medicine, Faculty of Medicine, Chulalongkorn University, Bangkok 10330, Thailand

**Keywords:** sepsis, cecal ligation and puncture, BAM15, uncoupling agent, extracellular flux

## Abstract

BAM15 (a mitochondrial uncoupling agent) was tested on cecal ligation and puncture (CLP) sepsis mice with in vitro experiments. BAM15 attenuated sepsis as indicated by survival, organ histology (kidneys and livers), spleen apoptosis (activated caspase 3), brain injury (SHIRPA score, serum s100β, serum miR370-3p, brain miR370-3p, brain TNF-α, and apoptosis), systemic inflammation (cytokines, cell-free DNA, endotoxemia, and bacteremia), and blood–brain barrier (BBB) damage (Evan’s blue dye and the presence of green fluorescent *E. coli* in brain after an oral administration). In parallel, brain miR arrays demonstrated miR370-3p at 24 h but not 120 h post-CLP, which was correlated with metabolic pathways. Either lipopolysaccharide (LPS) or TNF-α upregulated miR370-3p in PC12 (neuron cells). An activation by sepsis factors (LPS, TNF-α, or miR370-3p transfection) damaged mitochondria (fluorescent color staining) and reduced cell ATP, possibly through profound mitochondrial activity (extracellular flux analysis) that was attenuated by BAM15. In bone-marrow-derived macrophages, LPS caused mitochondrial injury, decreased cell ATP, enhanced glycolysis activity (extracellular flux analysis), and induced pro-inflammatory macrophages (*iNOS* and *IL-1β*) which were neutralized by BAM15. In conclusion, BAM15 attenuated sepsis through decreased mitochondrial damage, reduced neuronal miR370-3p upregulation, and induced anti-inflammatory macrophages. BAM15 is proposed to be used as an adjuvant therapy against sepsis hyperinflammation.

## 1. Introduction

Sepsis, a life-threatening response to a systemic infection, is a major health-care issue around the world, due to a complicated combination of immune homeostasis, inadequate tissue perfusion, and reactive oxygen species that result in mortality and multi-organ injury, especially in the brain, kidney, and liver [1,2]. Although the renal and hepatic damages in sepsis are frequently mentioned [3,4,5], studies on the brain impact of sepsis are fewer. One of the sepsis complications is sepsis-associated encephalopathy (SE), which is a diffuse brain dysfunction caused by sepsis without evidence of brain infection, metabolic disturbance (electrolyte imbalance and glycemic coma), or other causes of encephalopathy (uremia and hepatic failure) [6,7]. Because approximately 70% of sepsis patients exhibit some symptoms of SE, with a wide range of clinical manifestations from moderate (confusion, inattention, and focus deficits) to severe characteristics (deep coma), SE is considered the most frequent complication of sepsis [8,9]. Due to the clinical similarity to other causes of encephalopathy, diagnosis of SE is based on the exclusion of other possible causes without definitively established clinical criteria [7,10]. Data on the sepsis influence on brain cell energy status are still fewer, despite the previously mentioned SE pathophysiology, including endothelial injury, blood–brain barrier defect, inflammation, neuron signaling interference, and cell apoptosis [11]. In contrast, it is well known that sepsis microbial molecules, especially lipopolysaccharide (LPS; a major cell wall component of Gram-negative bacteria), induce an alteration in macrophages toward the pro-inflammatory direction, referred to as “M1 macrophage polarization”, partly through an enhanced glycolysis activity which is the main source of cell energy during cytokine production [12]. Due to the well-known correlation between macrophage plasticity (pro- vs. anti-inflammation) and cell energy [13], the interference in macrophage energy status, for example by 2-deoxy-D-glucose (2DG; a glycolysis activity inhibitor) or N5, N6-bis(2-Fluorophenyl) [1,2,5] oxadiazolo [3,4-b] pyrazine-5,6-diamine (BAM15) (a mitochondrial uncoupler agent), reduces the cell activities and attenuates sepsis severity [14].

However, the energy status of brain cells is different from macrophages, as neurons have high energy consumption when compared with other cells [15]. As determined by the whole-body oxygen uptake, metabolic consumption of the brain and other organs is 20% and 2%, respectively, leading to high sensitivity to the energy depletion of neurons compared with other cells [15]. As such, SE might be an impact of sepsis on neuron cells, and the intervention on neuron energy status might be interesting. Due to sepsis-induced defects of the blood–brain barrier (BBB; a highly selective semipermeable membrane between blood and brain), several molecules in the blood, including pathogen-associated molecular patterns (PAMPs), especially lipopolysaccharide (LPS; a major cell wall structure of Gram-negative bacteria—the most abundant gut pathogen [16,17]) and damage-associated molecular patterns (DAMPs), can easily pass through the BBB. Notably, the presence of LPS in serum during sepsis is possibly a result of Gram-negative bacteria in blood or the translocation of LPS from the gut into the blood circulation (gut translocation) [1]. The stimulation from several molecules during sepsis alters brain and neuron activities, including the production of several microRNAs (miRs; a small endogenous non-coding RNA), RNA, and proteins (such as cytokines). Among different brain-induced miRs during sepsis, miR370-3p is highly expressed in the brain and can be detected in serum, which has been proposed as a biomarker for SE status [18,19] and possibly is associated with cell energy [20,21,22]. Because (i) the energy status of different cell types might be important in sepsis [23], (ii) BAM15 attenuates sepsis in the LPS injection mouse model, partly through the manipulations in the energy status of hepatocytes and RAW264.7 cells (macrophages) [24], and (iii) miR370-3p is highly upregulated in brain cells during sepsis [18], we hypothesize that miR370-3p might be associated with brain cell energy and BAM15 might attenuate sepsis severity in multiple organs, including the brain, through the energy interference.

Hence, BAM15 was tested in the cecal ligation and puncture (CLP) sepsis mouse model with in vitro experiments on PC12 (a pheochromocytoma neuron cell line) and bone-marrow-derived macrophages.

## 2. Results

### 2.1. BAM15 Attenuated Sepsis Severity in Cecal Ligation and Puncture Mice in Several Organs (Kidneys, Livers, Spleens, and Brains)

To test the impact of BAM15 in sepsis mice, BAM15 (5 mg/kg/dose) was subcutaneously administered after CLP surgery and at 6 h post-CLP before the determination of several parameters. Accordingly, BAM15 attenuated sepsis severity as indicated by survival analysis (Figure 1A), renal injury (blood urea nitrogen, serum creatinine, and kidney histology) (Figure 1D–F and Figure 2A), liver damage (alanine transaminase and liver histology) (Figure 1G,H and Figure 2B), spleen apoptosis (Figure 1I and Figure 2C), encephalopathy-associated clinical manifestation (SHIRPA score) (Figure 1J), brain injury parameters in serum (s100β and miR370-3p) (Figure 1K,L) and in brain tissue (miR370-3p, TNF-α, and apoptosis) (Figure 1M–O and Figure 2D), but not white blood cells (WBC) and neutrophils in peripheral blood (Figure 1B,C). Likewise, BAM15 also attenuated other parameters of systemic inflammation, as indicated by serum cytokines (TNF-α, IL-6, and IL-10) and serum cell-free DNA (cf-DNA) (Figure 3A–D), which might be due to the reduction in endotoxemia and bacteremia (Figure 3E,F). Additionally, to demonstrate the brain barrier defect during severe sepsis, the Evan’s blue (EB) dye procedure and the administration of GFP-*E. coli* prior to CLP surgery was performed. With EB, EB staining in the brain of CLP mice was higher than control and partially attenuated by BAM15 treatment (Figure 4A). Likewise, GFP-*E. coli* were detectable in the brain of mice at 24 h post-CLP (GFP-*E. coli* were orally administered at 6 h before CLP) and BAM15 administration attenuated the abundance of bacteria in mouse brains (Figure 4B,C). The translocation of GFP-*E. coli* from the intestine into the brain of sepsis mice supports the defect on either gut permeability or the blood–brain barrier [25,26,27,28].

### 2.2. Sepsis Enhanced MiR370-3p in Mouse Brains, Possibly Due to the Activation by Endotoxin and TNF-α, Which Was Attenuated by BAM15

Because BAM15 attenuated sepsis encephalopathy (SHIRPA score) (Figure 1J) along with miR370-3p in both serum and brain tissue of mice (Figure 1L,M), miR370-3p might be associated with the pathogenesis of sepsis encephalopathy. Indeed, the miR arrays from brain tissue demonstrated an increase in miR370-3p in the brain of mice at 24 h post-CLP (severe sepsis) that became undetectable at 120 h post-CLP (sepsis survivors) (Figure 5A), supporting the possible correlation of miR370-3p with the pathophysiology of sepsis encephalopathy [18]. With the miRWalk tool, the integrated database from TargetScan, miRDB, and miRWalk, miR370-3p is significantly associated with several energy metabolism pathways that might induce brain cell dysfunctions, including glycolysis (carbohydrate, inositol phosphate, and pyrophosphate), mitochondrial activity (tricarboxylic acid cycle and electron transport chain), and lipid metabolism (Figure 5B), partly through the regulation of several enzymes (Figure 6). The binding energy of miR370-3p to the 3′UTR of the mRNA target in the metabolism pathway was also shown (Figure 6). Interestingly, the incubation of neuron cells (PC-12) with LPS and TNF-α (but not IL-6 and IL-10; data not shown) upregulated miR370-3p in neuron cells after 24 h and 48 h of the experiments, respectively (Figure 7A).

Due to (i) increased miR370-3p and TNF-α in sepsis mouse brains (Figure 1M,N and Figure 5A), (ii) TNF-α (and LPS) upregulated miR370-3p in neuron cells (Figure 7A), and (iii) the possible correlation of miR370-3p with cell energy status or the molecules that possibly correlated with energy pathways (Figure 5B and Figure 6), the elevated LPS and TNF-α in sepsis mice might upregulate miR370-3p in mouse brains that altered the neural functions through the interference on cell energy status. Hence, the evaluations on PC-12 cells after activation by LPS, TNF-α, or miR370-3p transfection with or without BAM15 were performed. Indeed, all of these activations (LPS, TNF-α, and miR370-3p transfection) similarly induced mitochondrial injury (decreased mitochondrial membrane potential; MMP) and reduced total PC12 cellular ATP (Figure 7B,C), possibly through the profound mitochondrial activities as indicated by the increased oxygen consumption rate (OCR). Meanwhile, LPS and TNF-α, but not miR370-3p transfection (Figure 7D–F), demonstrated a tendency of increased glycolysis activity as demonstrated by the graph of extracellular acidification rate (ECAR) when compared with control or BAM15 alone (Figure 7G–I) but did not reach a significant level in the calculated area under the curve (AUC) of ECAR (Figure 7L). The alteration of MMP, but not cellular ATP, was normalized by BAM15, as indicated by the OCR and ECAR graphs (Figure 7J–L) that were shifted toward the control group in the extracellular flux analysis (Figure 7B–L). In PC12 cells, miR370-3p transfection but not LPS and TNF-α reduced glycolysis activity, as evaluated by the AUC of ECAR (Figure 7L). Notably, BAM15 without other stimulations reduced mitochondrial activity but did not increase glycolysis (Figure 7L). Perhaps, the BAM15 anti-inflammatory property [29,30] prevents neuronal cell injury through the blockage of too profound pro-inflammation by reducing MMP and total ATP in neuron cells before being activated by any stimulators (Figure 7B–L).

### 2.3. BAM15 Reduced Pro-Inflammatory Macrophages, Partly through an Alteration on Cell Energy Status, Which Might Be Responsible for Sepsis Attenuation

Because of (i) the attenuation of systemic inflammation in sepsis mice by BAM15 (Figure 3A–C), (ii) the importance of macrophages on sepsis inflammation [29,31,32], and (iii) the correlation between cell energy and macrophage activities [33], the BAM15 anti-inflammatory property against sepsis might be due to an impact of BAM15 on macrophage energy status. As such, LPS or BAM15 alone without LPS reduced MMP and total cell ATP (Figure 8A,B), similar to PC12 (Figure 7B,C); however, the alteration of cell energy after the cell manipulation in the extracellular flux analysis was different from PC12 neuronal cells. Accordingly, LPS profoundly reduced mitochondrial activity, as indicated by the graph of OCR, including basal respiration, maximal respiration, and respiratory reserve, with an enhanced glycolysis activity (AUC of ECAR) (Figure 8C–G). Due to the dependence on glycolysis for the M1 macrophage cytokine production [34,35,36], the reduced glycolysis activity with mitochondrial neutralization by BAM15 shifted LPS-activated macrophages from M1 pro-inflammation toward M2-anti-inflammation (downregulated *TNF-α*, *IL-6*, *iNOS*, and *IL-1β*) (Figure 9A,B,D,E). However, BAM15 did not upregulate anti-inflammatory genes (*IL-10*, *Arginase-1*, *Fizz-1*, and *TGF-β*) (Figure 9C,F–H). Notably, LPS did not upregulate miR370-3p, and BAM15 alone without LPS did not alter the OCR and ECAR graph in macrophages (data not shown). While BAM15 protected LPS-induced neuronal cell injuries (PC12), possibly through the prevention of the overwhelming activities by decreasing mitochondrial activity (Figure 7B–L), BAM15 reduced macrophage responses against LPS through the shift of the cell energy status toward anti-inflammatory M2 polarization (Figure 8A–G and Figure 9A–H). These data supported the BAM15 anti-inflammatory effect through the different mechanisms in the specific cell types depending on the influences of the energy status of those cells.

## 3. Discussion

Cytokines (especially TNF-α) and LPS from sepsis upregulated miR370-3p in the brain, at least in part, facilitated encephalopathy through an alteration of energy status in neuron cells. BAM15 attenuated injury in several organs, including the brain, through the reduction in cell energy that decreased inflammatory responses in neuron cells and macrophages.

### 3.1. BAM15 Attenuated Encephalopathy and Systemic Inflammation in Sepsis via Downregulated miR370-3p

Earlier work shows that BAM15 attenuates sepsis in the LPS injection model [29]; however, CLP is a sepsis mouse model which more resembles human sepsis in terms of bacteremia, cytokine levels, and natural course of the disease [2]. Here, BAM15 improved survival, renal injury, liver damage, spleen apoptosis, and brain injury along with systemic inflammation (serum cytokines). Regarding brain damage, BAM15 attenuated encephalopathy score (SHIRPA) and several injury parameters, including brain cytokine (TNF-α), apoptosis (activated caspase 3), blood–brain barrier defect (Evan’s blue dye assay), s100β, and *miR370-3p*. Indeed, TNF-α is a key mediator of septic encephalopathy, as TNF-α-deficient mice are resistant to LPS-induced encephalopathy [37,38] and TNF-α might activate the TNF-related apoptosis-inducing ligand (TRAIL) receptor leading to cell apoptosis on neurons, astrocytes, and oligodendrocytes [39]. Moreover, the brain injury and cytokine activation might also be responsible for the damage to the BBB, which is the endothelial barrier between the blood circulation and brain cells [40,41,42]. Interestingly, the sepsis-induced BBB defect was severe enough to see the presence of GFP-*E. coli* (the viable bacteria) in mouse brains, and the BBB defect might also allow the translocation of the brain-producing molecules into the blood circulation [18]. Indeed, there was also an increase in s100β and miR370-3p in sepsis mouse brains that could be detected in serum and proposed as biomarkers for encephalopathy during sepsis [18,43]. While s100β is an astrocyte-specific calcium-binding protein with extensive study [44,45], the physiologic importance of miR370-3p is still unknown. Because miRs are stable in blood and are smaller than most proteins, they might be easier to pass through the BBB, leading to a more sensitive biomarker of brain disorder conditions [46,47,48,49,50]. Here, miR arrays from the brains of sepsis mice demonstrated miR370-3p upregulation at 24 h post-CLP (the peak sepsis injury) and spontaneously downregulated at 120 h post-CLP (sepsis survivor) when compared with 24 h post-sham control, supporting the association between miR370-3p and sepsis brain injury. Additionally, miR370-3p was associated with several pathways of energy metabolism, partly through the possible interaction between miR370-3p through several genes (Figure 5 and Figure 6). For example, miR370-3p was predicted to regulate the *ACADSB* (Acetyl CoA dehydrogenase short/branch chain) gene, which is crucial for lipid beta-oxidation in mitochondria [51]. Similarly, upregulated miR370-3p during sepsis may decrease mitochondrial metabolism through the impact of the *Mgll* (Monoaryl glycerol lipase) gene, which generates free-fatty acid and glycerol for mitochondria [52]. Taken together, miR370-3p is a sensitive marker to determine sepsis-induced brain injury and mediates cell metabolism during sepsis. BAM15-dependent miR370-3p downregulation might increase cell metabolism and increased cell viability. However, the conclusions are based on the sepsis model in the male gender, which is possibly more severe than the female sepsis [53]. The exploration of only male sepsis might limit the clinical translation from these results. Further studies are needed to characterize the role of miR370-3p in sepsis conditions.

With the in vitro experiments, the activation by LPS and TNF-α upregulated miR370-3p in the neuron cells (PC12) at 24 and 48 h post-stimulation, respectively, and all interventions (LPS, TNF-α, and miR370-3p transfection) similarly caused the mitochondrial injury (reduced MMP) and reduced total cell ATP, with enhanced mitochondrial activity (maximal respiration and respiratory reserve) but not glycolysis. Indeed, mitochondrial injury after profound neuronal cell activities is reported [54,55]. BAM15 normalized PC12 mitochondrial activity into levels of the control, which possibly attenuated sepsis-induced cell injury. Although BAM15 without all of the stimulations reduced MMP and total ATP in control PC12 cells, BAM15 did not show any adverse effects in the control mice. Prevention of the sepsis-induced overwhelming mitochondrial activations might be a major mechanism of the attenuation of sepsis encephalopathy by BAM15. In a normal situation, oxidative phosphorylation (OXPHOS) is an effective process of mitochondrial ATP production that is processed by the electrochemical proton gradient across the mitochondrial inner membrane through ATP synthase [56]. However, some protons do not enter the ATP synthesis process but leak back into the mitochondrial matrix, referred to as “mitochondrial uncoupling”, which leads to a decrease in proton gradients and ATP production [57]. The mitochondrial uncoupling agents, such as BAM15, have a capacity to transfer protons back to the process of ATP synthesis, leading to the enhanced ATP synthesis in a very short period of time followed by an MMP impairment that reduces the further ATP synthesis from the subsequent stimulations [58,59,60], including the pro-inflammatory responses [12,61,62]. Perhaps, the reduced MMP in neuron cells by BAM15 decreased the responses against LPS and cytokines (TNF-α) that limited sepsis-induced miR370-3p upregulation, resulting in less severe mitochondrial energy exhaustion, improved neuronal function, and reduced severity of sepsis encephalopathy.

### 3.2. BAM15 Induced Anti-Inflammatory Macrophages in Sepsis

Among several mechanisms that induce multi-organ injury in sepsis, systemic inflammation from hypercytokinemia is one of the important causes in which macrophages are important immune cells with prominent cytokine production properties [62,63,64,65]. While LPS enhanced mitochondrial activity without glycolysis alteration in PC12 cells, LPS reduced mitochondrial function and increased glycolysis in bone-marrow-derived macrophages that facilitated pro-inflammation, partly through glycolysis-associated pr-inflammation in macrophages [66,67]. Perhaps, the responses against pathogen molecules of neuron cells and macrophages are different, as macrophages (or microglia in the brain) are better equipped with several pattern recognition receptors when compared with parenchymal cells in most organs [68,69]. Despite the different impact of LPS on PC12 and macrophages in the extracellular flux analysis, LPS induced mitochondrial injury (reduced MMP), and decreased total cell ATP in both cell types (PC12 and macrophages). However, BAM15 attenuated mitochondrial injury and decreased pro-inflammatory responses in both cell types, which partly explained the sepsis attenuation effect of BAM15. The reduced cell energy status by BAM15 to the level that has a limited response against inflammatory stimulators might protect the cells from cell injury from the too profound responses, for example, excessive ROS, apoptosis induction, -->excessive ROS (apoptosis induction) [70,71,72]. Although systemic inflammation in sepsis is the main cause of multi-organ injury in sepsis, the overall sepsis attenuation property of BAM15 is possibly not only explained through the systemic anti-inflammation but also by the direct effect of BAM15 on mitochondrial protection and cell energy status alteration in the different specific cells in all organs. Indeed, the direct protective effects of BAM15 are demonstrated in hepatocytes [29] and renal tubular cells [73]. Here, we propose to use BAM15 as adjuvant therapy in sepsis, especially with encephalopathy complications. Hence, we hypothesized that sepsis molecules (LPS, cytokines, and some viable bacteria) pass through the damaged BBB and upregulate miR370-3p in brain cells, partly through the activation of TNF-α receptor (TNFR) and TLR-4 by TNF-α and LPS, respectively, that enhance mitochondrial injury (Figure 10). In blood, these sepsis molecules induce pro-inflammatory M1 macrophage polarization with mitochondrial injury, and the inflammation in either brains or macrophages is attenuated by the mitochondrial protection property of BAM15 (Figure 10). More studies on this topic are warranted.

In conclusion, upregulation of miR370-3p in sepsis encephalopathy was a result of the activation by sepsis molecules (LPS and TNF-α) which was attenuated by BAM15 along with decreased organ damage through BAM-15-induced anti-inflammatory macrophages. A strategy of sepsis adjuvant therapy through interference on cell energy status is interesting.

## 4. Materials and Methods

### 4.1. Animal and Animal Model

The animal protocol (029/2561) was approved by the Institutional Animal Care and Use Committee of the Faculty of Medicine, Chulalongkorn University, following the US National Institutes of Health (NIH) animal care and use protocol. Male 8-week-old mice weighing 20–22 g from Nomura Siam (Pathumwan, Bangkok, Thailand) were used. Only male mice were used in the experiments due to the well-known characterization of sepsis in male mice from our group, the effect of gender difference in sepsis should be considered for the clinical translation [53]. The mice were kept in conventional clear plastic cages with free access to water and food (SmartHeart Rodent; Perfect companion pet care, Bangkok, Thailand) with a 12:12 h light/dark cycle at 22 ± 2 °C and 50% relative humidity. Cecal ligation and puncture (CLP) procedures were performed through an abdominal incision under isoflurane anesthesia following the previous publications with the ligation at 10 mm from the cecal tip and punctured twice with a 21-gauge needle [33,74,75,76,77]. In a sham operation, the cecum was identified before closing the abdomen. Additionally, BAM15, a mitochondrial protonophore uncoupler agent, purchased from Sigma-Aldrich (St. Louis, MO, USA) at 5 mg/kg/dose in 3% dimethyl sulfoxide (DMSO) or DMSO alone (control group) was intravenously administered (tail vein) before CLP surgery and subcutaneous injection at 6 h post-CLP. Fentanyl at 0.03 mg/kg in 0.5 mL of normal saline solution (NSS) was subcutaneously administered for analgesia and post-operative fluid replacement. To determine the severity of sepsis encephalopathy, SHIRPA score (SmithKline Beecham, Harwell, Imperial College, Royal London Hospital, phenotype assessment) was used as previously described [18]. As such, SHIRPA score is based on several parameters, such as mouse position, respiration, activity, reflex responses, body tone, etc., and is designed for the evaluation of several mouse aspects (behavioral, neurological, and physiological characteristics) [78] and is adapted to use for encephalopathy [18,79,80,81]. Mice were sacrificed by cardiac puncture under isoflurane anesthesia with the sample collection (internal organs and blood). Organs and serum were kept at −80 °C until analyzed. Additionally, the brain tissue was prepared by RNAlater (Thermo Fisher Scientific, Waltham, MA, USA) with homogenization using Qiazol lysis (QIAGEN, Hilden, Germany) for microRNA (miR) evaluation (see miR measurement).

### 4.2. Mouse Serum Sample Analysis

For total peripheral blood leukocytes, blood was mixed with 3% volume by volume (*v*/*v*) of acetic acid for red blood cell lysis in a ratio of blood and acetic acid at 1:20 by volume before counting with a hemocytometer. In parallel, the Wright-stained blood smears were determined for the percentage of neutrophils. Kidney injury was determined by QuantiChrom Urea Assay (DUR2-100) and QuantiChrom Creatinine Assay (DICT-500) (BioAssay, Hayward, CA, USA), while liver damage was evaluated by EnzyChrom ALT assay (EALT-100, BioAssay). Serum cytokines (TNF-α, IL-6, and IL-10) and serum endotoxin (LPS) were measured with ELISA (eBioscience, San Diego, CA, USA) and the Limulus Amebocyte lysate test (Associates of Cape Cod, East Falmouth, MA, USA), respectively. The values of LPS < 0.01 EU/mL were recorded as 0 due to the limitation of the standard curve. Blood in serial dilutions was directly spread onto blood agar (Oxoid, Hampshire, UK) and incubated at 37 °C for 24 h before colony enumeration. For cell-free DNA (cf-DNA), the DNA in serum was extracted with 5 M potassium acetate/acetic acid buffer and quantified by a Nanodrop 100 spectrophotometer (NanoDrop 3300; Thermo Scientific, Wilmington, DE, USA). To determine sepsis-induced encephalopathy, S100β and miR370-3p in serum were measured by ELISA assay (SEA567Mu, Cloud-Clone Corp., Katy, TX, USA) and miR determination (see miR measurement), respectively, following a previous publication [18]. 

### 4.3. Blood–Brain Barrier Permeability Analysis (Evan’s Blue Dye and GFP-E. coli) 

The analysis of blood–brain barrier (BBB) permeability was performed by (i) the Evan’s blue dye (EB) assay for detection of microvascular leakage and (ii) the oral administration of green-fluorescent-producing Escherichia coli (GFP-*E. coli*) prior to CLP surgery. For the EB procedure, 1% EB (Sigma-Aldrich) at 2 mL/kg in 0.9% sodium chloride was administered via tail vein at 30 min before sacrifice following a published procedure [18]. At sacrifice, phosphate buffer solution (PBS) was perfused through the left ventricle until the blue color in the blood was totally eliminated and brains were weighed, snap-frozen in liquid nitrogen, homogenized in formamide (Sigma-Aldrich, St. Louis, MO, USA) in a ratio of the brain: formamide at 0.4 mg: 1 mL at 55 °C for 18 h before centrifugation, and EB in the supernatant was measured with the absorbance at 620 nm in comparison with the EB standard quantitative curve. Because all Evan’s blue dye is virtually bound to albumin and serum albumin cannot cross the BBB, the presence of EB in the neural tissue implies the BBB defect. In parallel, to track bacterial dissemination in the brain, GFP-*E. coli* (25922GFP) from American Type Culture Collection (ATCC, Manassas, VA, USA) at 1 × 10^9^ CFUs in 0.3 mL PBS was orally administered at 3 h before CLP surgery and fluorescent intensity in the internal organs was detected using ZEISS LSM 800 (Carl Zeiss, Oberkochen, Germany) as previously described [82,83].

### 4.4. Histology and Tissue Cytokines

Histology on Hematoxylin and Eosin (H&E) staining at 200× magnification was semi-quantitatively evaluated. For the kidney injury score, the injury score was defined by the area of injury (tubular epithelial swelling, loss of brush border, vacuolar degeneration, necrotic tubules, cast formation, and desquamation) using the following score: 0, area < 5%; 1, area 5–10%; 2, area 10–25%; 3, area 25–50%; 4, area > 50% [12,18,28,61]. The liver histological score was evaluated in accordance with a previous publication [84,85]. Accordingly, a combination of scores from the characteristics of hepatocyte injury based on cytoplasmic color fading, vacuolization, nuclear condensation, nuclear fragmentation, nuclear fading, and erythrocyte stasis ranging from 0 to 5 was multiplied by grades of the damage: 0, no injury; 1, mild injury; 2, moderate injury; 3, severe injury represented liver injury score. For apoptosis detection in the spleen and brain, 4 mm thick paraffin-embedded mouse organs after 10% formalin fixation were stained by anti-active caspase 3 antibody (Cell Signaling Technology, Beverly, MA, USA), detected by immunohistochemistry, and expressed in positive cells per high-power field (200× magnification).

### 4.5. MicroRNA Measurement

Total RNAs of serum and tissue samples were extracted using QIAGEN miRNeasy serum/plasma kit (QIAGEN, Hilden, Germany) according to the manufacturer’s instructions. The microRNAs (miRs) were converted to cDNA by TaqMan™ MicroRNA Assays kit (Applied Biosystems, Waltham, MA, USA) in RT-PCR (reverse transcription-polymerase chain reaction) machine (SimpliAmp™ Thermal Cycler systems, Applied Biosystems) using the mmu-miR370-3p primer (ID 002275) (Thermo Fisher Scientific) and cDNA samples for quantitative PCR through TaqMan™ Universal PCR Master Mix (Applied Biosystems, Waltham, MA, USA). Relative expression was calculated using the ∆∆CT method and normalized to the expression of snoRNA202 and cel-miR-39 for tissue and serum samples, respectively (Applied Biosystems). Because the miR results could be interfered by cell contamination and red blood cell rupture [86], sera were collected after centrifuged at 5000× rpm for 10 min before determination of hemolysis and RNA purity. Then, sample discoloration was excluded to reduce the interference from hemolysis. The ratio of absorbance at 260 nm and 280 nm is used to assess the purity of RNA, and the RNA samples with an absorbance ratio (absorbance at 260/absorbance at 280) higher than 2.0 were used.

### 4.6. MiR Sequencing Analysis

MicroRNA profiling from miR arrays of mouse brains with sepsis (at 24 h post-CLP), sepsis survivors (at 120 h post-CLP), and sham control was performed to explore the expression of different miRs in sepsis brains according to a previous publication [18]. Briefly, the brain tissue was prepared by QIAGEN miRNeasy kit (QIAGEN), and the libraries started with 1 µg total RNA for each sample. Then, the samples were processed with the miRNA array of BGISEQ-500platform and subsequently processed in R-Bioconductor (library package EdgeR) that were normalized by the trimmed mean of M-value between each pair of samples. ANOVA analysis was conducted and the *p*-value less than 0.05 as corrected with Bonferroni’s method was determined as a statistical significance. To understand the probable functions of mmu-miR370-3p (miR370-3p), the putative target was predicted using miRWalk 2.0 [87], which allows the merging of the predicted information from TargetScan and miRDB database. Additionally, the related pathways of the presumed target genes were demonstrated using the Reactome database [88], which corrected the *p*-value, and false discovery rate (FDR) technique, less than 0.05. MGI gene names were used according to ENSEMBL database gene ID.

### 4.7. Experiments in a Neuron Cell Line

A neuron cell line of PC-12 (ATCC CRL-1721) was incubated in complete media (RPMI1640, ATCC) with 10% *v*/*v* Horse Serum (HS), 5% *v*/*v* fetal bovine serum, and 1% *v*/*v* Penicillin-Streptomycin (Thermo Fisher Scientific) for 24 h before starting the experiment. Then, PC-12 at 1 × 10^5^ cells/well was incubated for 48 h with recombinant mouse TNF-α (100 ng/mL) (Sigma-Aldrich) or lipopolysaccharide (LPS) from Escherichia coli 026: B6 LPS (Sigma-Aldrich) at dose 1 mg/well or miR370-3p transfection (48 h) with or without BAM15 (at 10 nM/ well) before evaluation of extracellular flux analysis of cell energy status (as mentioned below), mitochondrial abundance (Mitotracker green fluorescent assay), and total cellular ATP. As such, mitochondrial membrane potential (MMP; mitochondrial function) was determined by MitoTracker, using 200 nM of Mitotracker red CMXRos (Molecular Probes Inc., Eugene, OR, USA), that was incubated at 37 °C for 15 min before fixing with cold methanol at −20 °C and measured by microplate reader at excitation OD579 nm and emission OD599 nm as previously described [84]. In parallel, cellular ATP analysis was performed by luminescent ATP detection assay (Abcam, Cambridge, UK) according to the manufacturer’s protocol [84]. For the transfection of miR 370-3p, has-miR370-3p (Ambion Inc., Thermo Fisher Scientific, Waltham, MA, USA) or miRNA-negative control (Ambion) (5 µL) was mixed with OptiMEM I (Invitrogen, Waltham, MA, USA) (final concentration of 100 nM in 50 µL RNAimax) for 10 min before mixing with lipofectamine 2000 (100 µL) in 100 µL OptiMEM I on a shaker at room temperature for 40 min. Then, the mixture was incubated with PC-12 at 5 × 10^5^ cells/well in 5% CO_2_ at 37 °C before retrieving the cells for further experiments.

### 4.8. Macrophage Experiments

Bone-marrow-derived macrophages were prepared from the healthy mice as previously described [26,28,89,90,91] using femurs and tibias of mice. Briefly, the bone marrow was collected by 6000 rpm centrifugation at 4 °C and incubated for 7 days with modified Dulbecco’s modified Eagle medium (DMEM) with conditioned media of the L929 cell line, containing macrophage-colony stimulating factor, in a humidified 5% CO2 incubator at 37 °C. Then, LPS (*E. coli* 026: B6, Sigma-Aldrich) at 100 ng/mL with or without 10 nM BAM15 or media control alone (DMEM) was incubated with macrophages at 1 × 10^5^ cells/well at 37 °C for 24 h before the preparation of total RNA using Trizol, quantified by a Nanodrop ND-1000 (Thermo Fisher Scientific), converted into cDNA by the Reverse Transcription System, and performed real-time quantitative reverse transcription-polymerase chain reaction (qRT-PCR) using the SYBR Green system (Applied biosystem, Foster City, CA, USA) for the expression of several genes. Relative expression was calculated based on the ΔΔCT method (2^−∆∆CT^) relative to the *β-actin* housekeeping gene. Primers for cytokines (*TNF-α*, *IL-6*, and *IL-10*), M1 pro-inflammatory macrophage polarization (*iNOS* and *IL-1β*), and M2 anti-inflammatory macrophage polarization (*Fizz-1*, *Arginase-1*, and *TGF-β*) were used (Table 1). In parallel, mitochondrial membrane potential using MitoTracker, using Mitotracker red CMXRos (Molecular Probes) (described above) [84], and extracellular flux analysis (described below) were also determined.

### 4.9. Extracellular Flux Analysis

Extracellular flux analysis with Seahorse XFp Analyzers (Agilent, Santa Clara, CA, USA) was used to determine the energy status of the cells, with oxygen consumption rate (OCR) and extracellular acidification rate (ECAR) representing mitochondrial function (respiration) and glycolysis activity, respectively [92]. For OCR evaluation, the stimulated macrophages at 1 × 10^5^ cells/well were incubated for 1 h in Seahorse media (DMEM complemented with glucose, pyruvate, and L-glutamine) (Agilent, 103575–100) before activation by different metabolic interference compounds such as oligomycin, carbonyl cyanide-4-(trifluoromethoxy)-phenylhydrazone (FCCP), and rotenone/antimycin A. The respiratory data of mitochondrial function were analyzed by Seahorse Wave 2.6 software based on the following equations: respiratory capacity (maximal respiration) = OCR between FCCP and rotenone/antimycin A—OCR after rotenone/antimycin A and respiratory reserve = OCR between FCCP and rotenone/antimycin A—OCR before oligomycin. In parallel, glycolysis stress tests were calculated from the mitochondrial stress test using the wave program of Seahorse XF Analyzers (Agilent) and demonstrated by the area under the curve of the ECAR graph as calculated by the trapezoidal rule [93].

### 4.10. Statistical Analysis

All data were analyzed by Statistical Package for Social Sciences software (SPSS 22.0, SPSS Inc., Chicago, IL, USA) and Graph Pad Prism version 7.0 software (La Jolla, CA, USA). Results were presented as mean ± standard deviation (SD). The differences between multiple groups were examined for statistical significance by one-way analysis of variance (ANOVA) with Tukey’s analysis. The survival analysis and time-point data were determined by the Log-rank test and repeated measures ANOVA, respectively. A *p*-value < 0.05 was considered statistically significant.

## Figures and Tables

**Figure 1 ijms-23-05445-f001:**
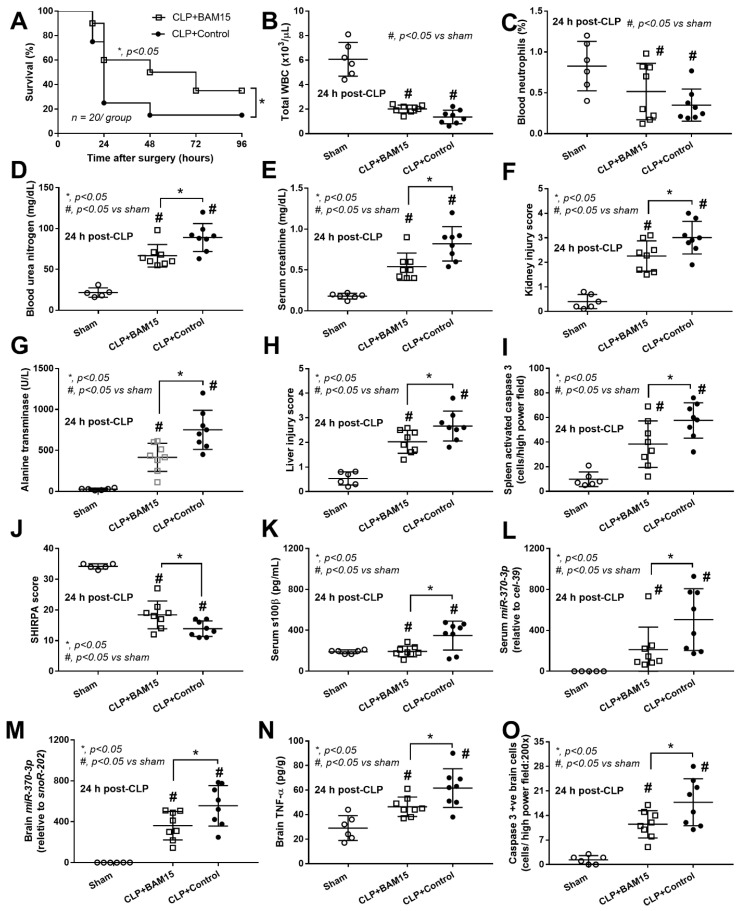
Characteristics of cecal ligation and puncture (CLP) sepsis or sham mice at 24 h after surgery with BAM15 or vehicle (control) as evaluated by survival analysis (*n* = 20/group) (**A**), total white blood cell (WBC) and neutrophils in peripheral blood (**B**,**C**), renal function (blood urea nitrogen, serum creatinine, and kidney histological score) (**D**–**F**), liver injury (alanine transaminase and liver histological score) (**G**,**H**), spleen apoptosis (activated caspase 3) (**I**), encephalopathy clinical score (SHIRPA score; see Methods) (**J**), and encephalopathy parameters in serum (S100β and miR370-3p) (**K**,**L**), brain miR370-3p, brain TNF-α, and brain apoptosis (activated caspase 3) (**M**–**O**) are shown (*n* = 6–8/group). #, *p* < 0.05 vs. sham; * *p* < 0.05 vs. the indicated groups as determined by ANOVA with Tukey’s analysis. The survival analysis is calculated by Log-rank test. Data from sham mice with BAM15 are not demonstrated due to the non-significant difference to the sham group.

**Figure 2 ijms-23-05445-f002:**
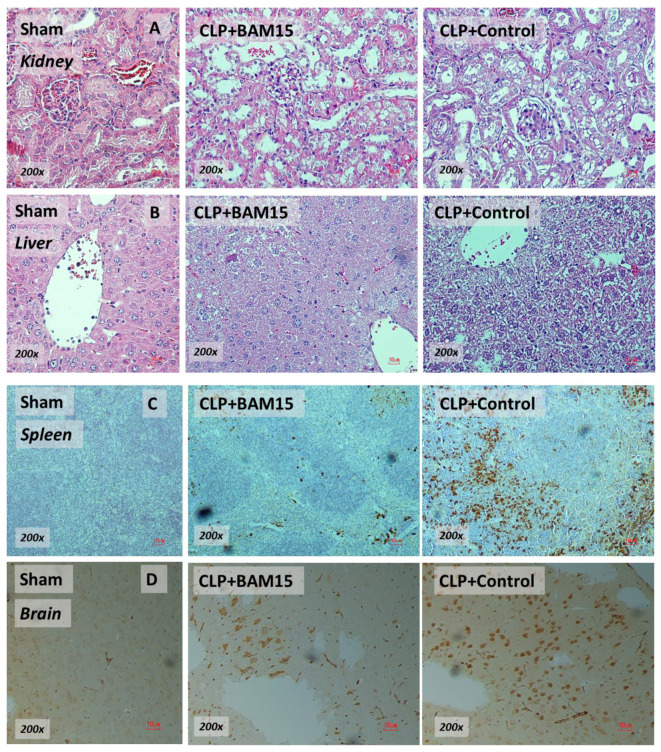
Representative pictures of Hematoxylin and Eosin (H&E) staining (kidneys and livers) (**A**,**B**) and activated caspase 3 immunohistochemistry (apoptosis) (spleen and brain) (**C**,**D**) of cecal ligation and puncture (CLP) sepsis or sham mice at 24 h after surgery with BAM15 or vehicle (control) are demonstrated. The pictures from sham mice with BAM15 are not shown due to the non-significant difference to the sham group.

**Figure 3 ijms-23-05445-f003:**
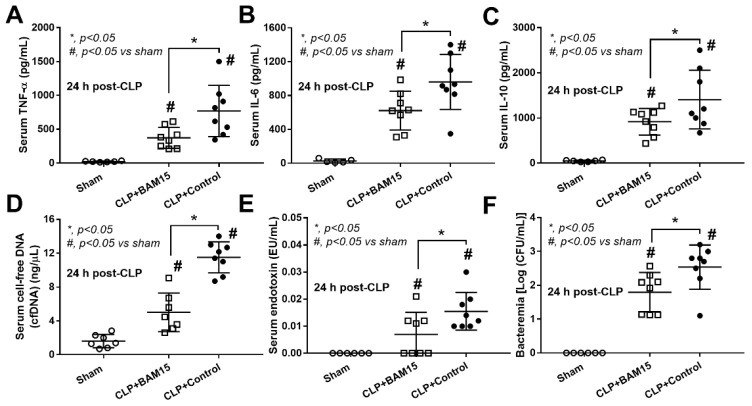
Characteristics of cecal ligation and puncture (CLP) sepsis or sham mice at 24 h after surgery with BAM15 or vehicle (control) as evaluated by serum cytokines (TNF-α, IL-6, and IL-10) (**A**–**C**), serum cell-free DNA (cf-DNA) (**D**), serum endotoxin (**E**), and bacteremia (**F**) are demonstrated (*n* = 6–8/group). #, *p* < 0.05 vs. sham; *, *p* < 0.05 vs. the indicated groups as determined by ANOVA with Tukey’s analysis. Data from sham mice with BAM15 are not demonstrated due to the non-significant difference to the sham group.

**Figure 4 ijms-23-05445-f004:**
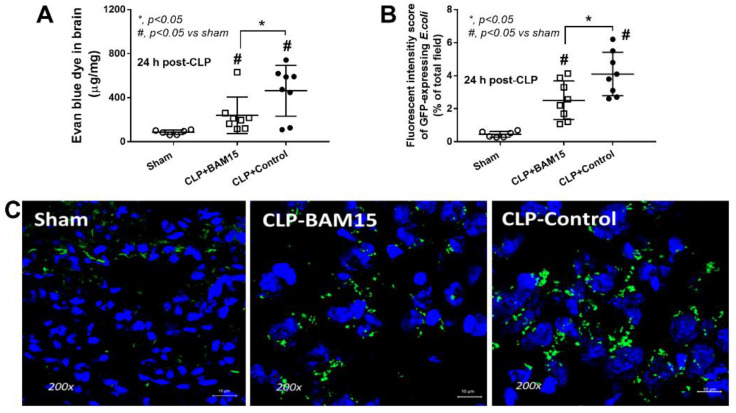
The abnormal permeability in the brain (Evan’s blue dye assay) (**A**) and abundance (fluorescent intensity) of green-fluorescent-producing (GFP) *E. coli* in the mouse brains at 24 h post-surgery with oral GFP-*E. coli* administration at 6 h prior to the operation (sham or CLP) with the representative fluorescent figures (**B**,**C**) are demonstrated (*n* = 6–8/group). #, *p* < 0.05 vs. Sham; *, *p* < 0.05 vs. the indicated groups as determined by ANOVA with Tukey’s analysis. Data from sham mice with BAM15 are not demonstrated due to the non-significant difference to the sham group.

**Figure 5 ijms-23-05445-f005:**
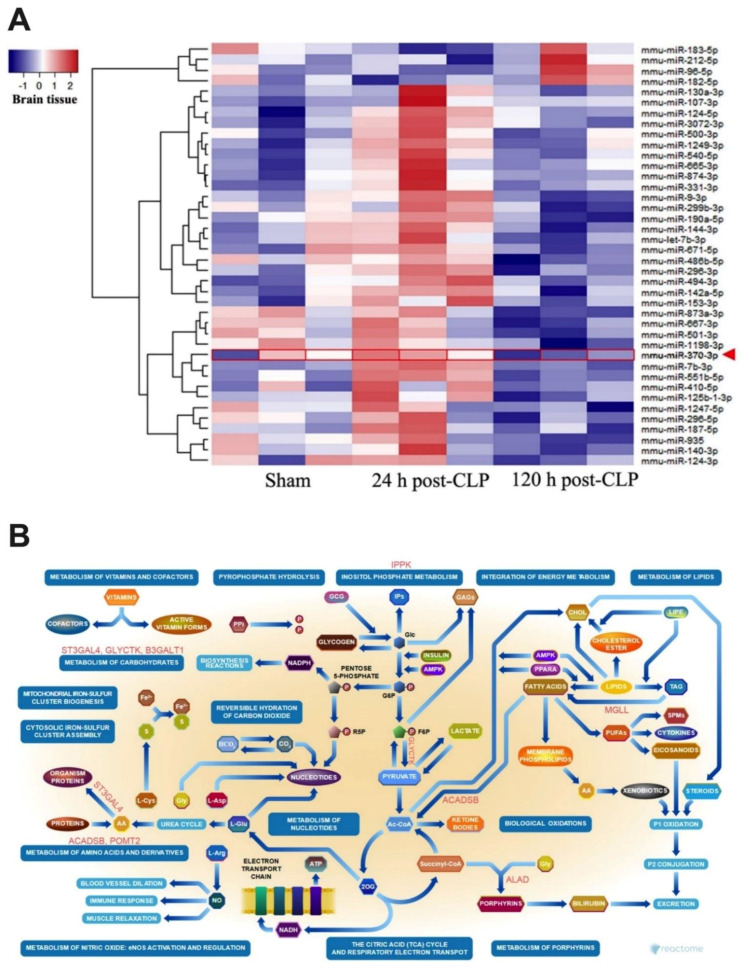
Heatmap illustration of microRNA arrays from the brains of mice with 24 h post-sham (sham), 24 h post-cecal ligation and puncture (CLP), and 120 h post-CLP (120 h post-sham group is not performed due to the similarity to 24 h post-sham) (**A**) (*n* = 3/group) and the predicted pathways that are associated with miR370-3p from the Reactome database are demonstrated (**B**). The arrowhead indicates miR370-3p on the analysis. The letters in red color are the predicted genes that are associated with miR370-3p from the Reactome analysis (more details on Figure 6) (This picture was provided by Biorender.com (accessed on 5 January 2022)).

**Figure 6 ijms-23-05445-f006:**
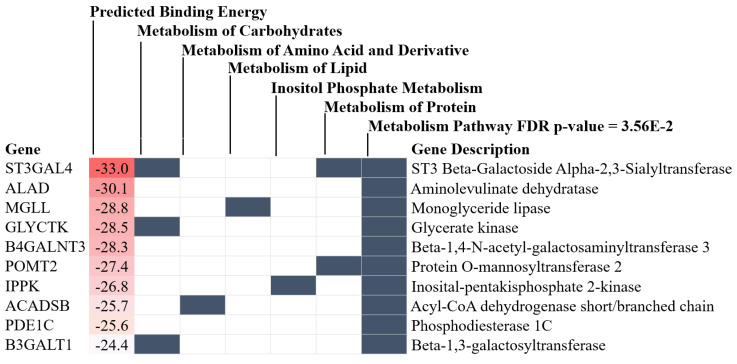
The figure shows binding energy of miR370-3p to the possible target in the metabolic pathway (Reactome analysis).

**Figure 7 ijms-23-05445-f007:**
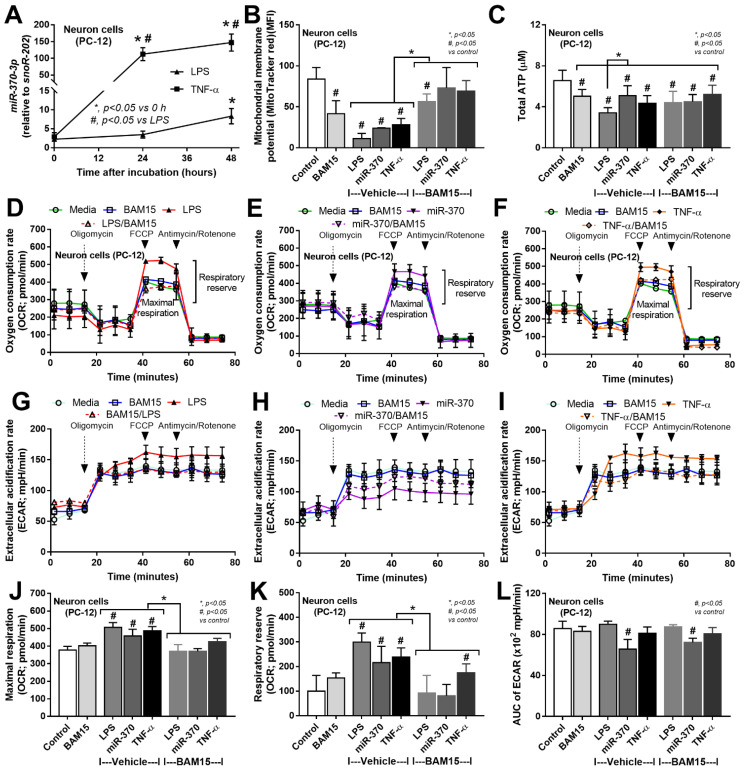
The expression of miR370-3p in neuron cells (PC-12 cell line) after stimulation with lipopolysaccharide (LPS) and TNF-α (**A**) and characteristics of PC-12 cells after 48 h stimulation by LPS, miR370-3p transfection, or TNF-α with or without BAM15, in comparison with control media (media) or BAM15 alone, as evaluated by mitochondrial membrane potential (**B**), total cell ATP (**C**), cell energy status through mitochondria (oxygen consumption rate; OCR) and glycolysis (extracellular acidification rate; ECAR) with several parameters (maximal respiration, respiratory reserve, area under the curve (AUC) of ECAR) (**D**–**L**) are also demonstrated. #, *p* < 0.05 vs. control; *, *p* < 0.05 vs. the indicated groups. The statistical analysis was determined by repeated measure ANOVA (time-point data) and ANOVA with Tukey’s method. #, *p* < 0.05 vs. control; independent triplicate experiments were performed. Notably, media and BAM15 control groups of miR370-3p transfection were also transfected by miR370-3p mimic (miR-negative control). MFI, mean fluorescent intensity.

**Figure 8 ijms-23-05445-f008:**
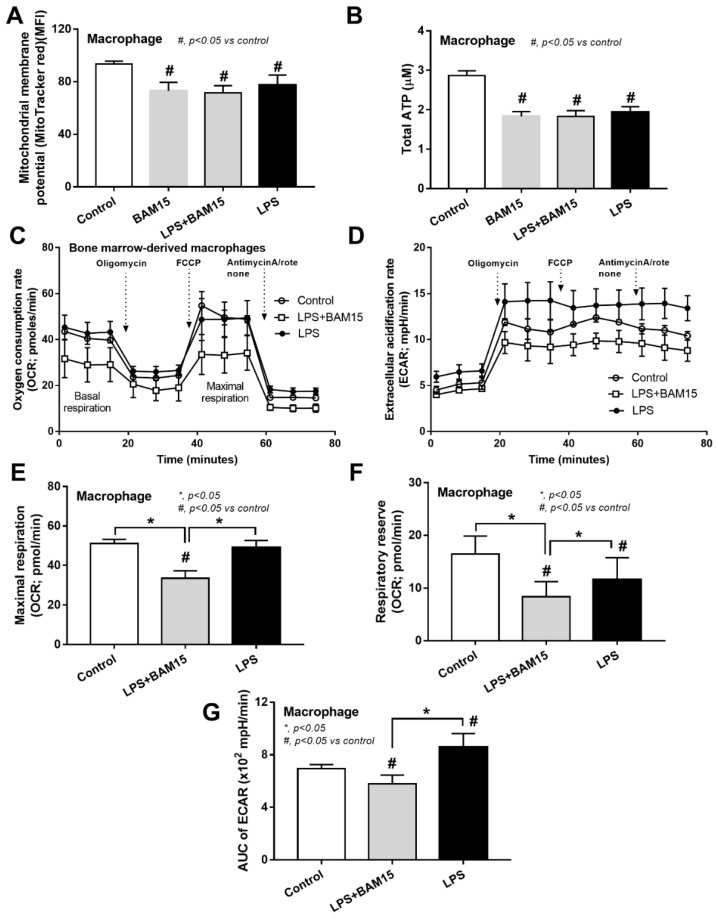
Characteristics of bone-marrow-derived macrophages after 24 h stimulation by LPS, with or without BAM15, in comparison with control media (media), as evaluated by cell energy status through mitochondrial membrane potential (**A**), total cell ATP (**B**), and extracellular flux analysis using mitochondria (oxygen consumption rate; OCR) and glycolysis (extracellular acidification rate; ECAR) with several parameters (basal respiration, maximal respiration, respiratory reserve, and area under the curve (AUC) of ECAR) (**C**–**G**) are demonstrated. The data from macrophages with BAM15 without LPS are not demonstrated in (**C**–**G**) due to the non-difference from the control group. #, *p* < 0.05 vs. control; *, *p* < 0.05 vs. the indicated groups as determined by ANOVA with Tukey’s analysis. #, *p* < 0.05 vs. control; independent triplicate experiments were performed. MFI, mean fluorescent intensity.

**Figure 9 ijms-23-05445-f009:**
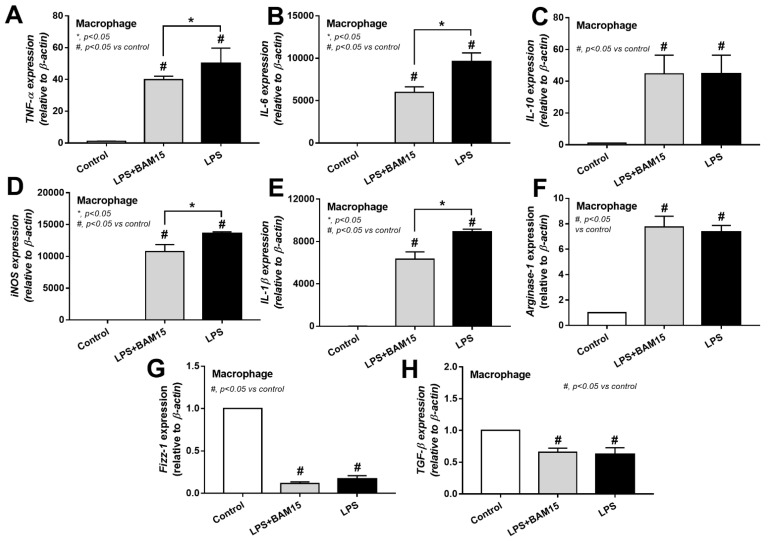
Characteristics of bone-marrow-derived macrophages after 24 h stimulation by LPS, with or without BAM15, in comparison with control media (media), as evaluated by gene expression of inflammatory cytokines (*TNF-α*, *IL-6*, and *IL-10*) (**A**–**C**), M1 macrophage polarization (pro-inflammation) (*iNOS* and *IL-1β*) (**D**,**E**), and M2 macrophage polarization (anti-inflammation) *(Arginase-1*, *Fizz-1*, and *TGF-β*) (**F**–**H**) are demonstrated. The data from macrophages with BAM15 without LPS are not demonstrated due to the non-difference from control group. #, *p* < 0.05 vs. control; *, *p* < 0.05 vs. the indicated groups as determined by ANOVA with Tukey’s analysis. #, *p* < 0.05 vs. control; independent triplicate experiments were performed.

**Figure 10 ijms-23-05445-f010:**
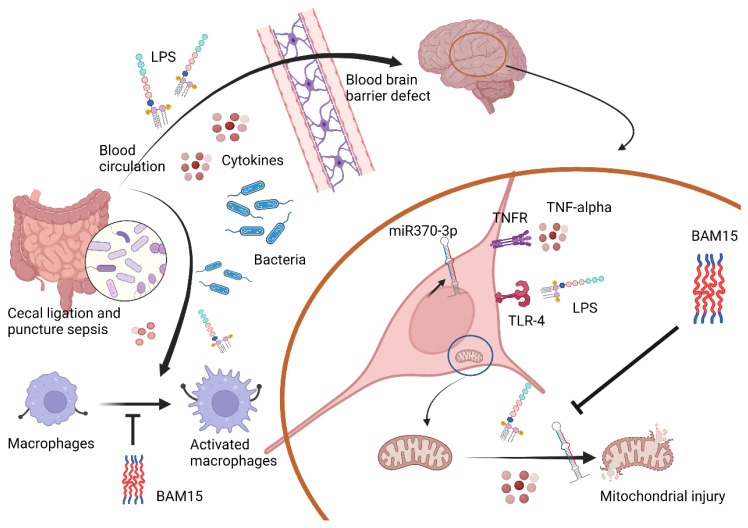
The proposed working hypothesis demonstrates the impacts of bacterial sepsis on brain cells and macrophages. In the brain, blood–brain barrier (BBB) defects in sepsis facilitate the translocation of several molecules, including cytokines, lipopolysaccharide (LPS), and bacteria into the brain. LPS and cytokines (especially TNF-α) upregulated miR370-3p through TLR-4 and TNF receptor (TNFR), respectively, that vigorously activate mitochondria leading to mitochondrial injury. In macrophages, several molecules in the blood during sepsis also activate macrophages, causing hyper-inflammatory responses. However, BAM15 blocks mitochondrial over-activity, which is beneficial for both brain cells and macrophages, resulting in less severe sepsis and sepsis encephalopathy.

**Table 1 ijms-23-05445-t001:** List of primers used in the study.

Primers	Forward	Reverse
Tumor necrosis factor-alpha (*TNF-α*)	5′ -CCTCACACTCAGATCATCTTCTC- 3′	5′ -AGATCCATGCCGTTGGCCAG- 3′
Interleukin-6 (*IL-6*)	5′ -TACCACTTCACAAGTCGGAGGC- 3′	5′ -CTGCAAGTGCATCATCGTTGTTC- 3′
Interleukin-10 (*IL-10*)	5′ -GCTCTTACTGACTGGCATGAG- 3′	5′ -CGCAGCTCTAGGAGCATGTG- 3′
Inducible nitric oxide synthase (*iNOS*)	5′ -ACCCACATCTGGCAGAATGAG- 3′	5′ -AGCCATGACCTTTCGCATTAG- 3′
Interleukin-1ß (*IL-1ß*)	5′ -GAAATGCCACCTTTTGACAGTG- 3′	5′ -TGGATGCTCTCATCAGGACAG- 3′
Arginase-1 (*Arg-1*)	5′ -CTTGGCTTGCTTCGGAACTC- 3′	5′ -GGAGAAGGCGTTTGCTTAGTTC- 3′
Transforming Growth Factor-β (*TGF-β*)	5′ -CAGAGCTGCGCTTGCAGAG- 3′	5′ -GTCAGCAGCCGGTTACCAAG- 3′
Resistin-like molecule-α (*FIZZ-1*)	5′ -GCCAGGTCCTGGAACCTTTC- 3′	5′ -GGAGCAGGGAGATGCAGATGA- 3′

## Data Availability

Data are contained within the article.

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
