# Peer review of "Sepsis Encephalopathy Is Partly Mediated by miR370-3p-Induced Mitochondrial Injury but Attenuated by BAM15 in Cecal Ligation and Puncture Sepsis Male Mice"

_ijms, 2022, doi:10.3390/ijms23105445_

Round 1
Reviewer 1 Report
Vary good manuscript on "Sepsis Encephalopathy is Partly Mediated by miR370-3p Induced Mitochondrial Injury but Attenuated by BAM15 in Cecal ligation and Puncture Sepsis Mice" I recommend publication of this manuscript.
Minor spelling and grammatical errors have to fix for publication.
Author Response
Reviewer 2
Main point
Despite recognition of the importance of sex and gender in most areas of research, important knowledge gaps persist owing to the general orientation of scientific attention to one sex category. The gap in the representation of females in studies on human subjects, but also in animal models has been well documented. (ok)
ANS: We thank the reviewer for the comment. We usually use male mice in sepsis just because of our habituate protocol and there was a possible less severe sepsis in female mice. We will note this limitation in several parts of our manuscript.
Title and abstract. If only one sex is included in the study, the title and the abstract should specify the sex of animals. Authors should indicate if and why the study model was based on one sex. The article’s title should specify this fact by including “in males” in the title and abstract. (ok)
ANS: We agree with the reviewer and put “male” in the title as following “Sepsis encephalopathy is partly mediated by miR370-3p induced mitochondrial injury but attenuated by BAM15 in cecal ligation and puncture sepsis male mice”.
Methods. Authors should justify the reasons for any exclusion of female mice. Methodological choices about sex in relation to your study should be reported and justified in the same way as other methodological choices. (ok)
ANS: We thank the reviewer for the comment and put a remark in this aspect as following “Only male mice were used in the experiments due to the well-known characterization of sepsis in male mice from our group, the effect of gender difference in sepsis should be considered for the clinical translation.”.
Discussion Authors should further discuss the implications of the lack of female mice analysis on the interpretation of the results. (ok)
ANS: We thank the reviewer for the comment and discuss this limitation as following “However, the conclusions are based on the sepsis model in the male gender which is possibly more severe than the female sepsis. The exploration of only male sepsis might limit the clinical translation from these results. Further studies are needed to characterize the role of miR-370-3p in sepsis conditions.”
Minor points
Materials and methods:
Page 17 Line 526: the sentence “The cDNA template and target primers based on the ΔΔCT method (2-∆∆Ct) relative to the β-actin housekeeping gene were conducted” is incorrect. Maybe you meant, “Relative expression was calculated based on the ΔΔCT method (2-∆∆Ct) relative to the β-actin housekeeping gene”. (ok)
ANS: We apologize for our mistake and correct it accordingly.
Moreover, Figure 9 shows gene expression of what? 2-∆∆Ct respect to the control? However, controls are not =1 in the graphs. (ok)
ANS: We apologize for our mistake and correct figure 9.
For the analysis of miRNA in serum, the authors should describe details of the methods used to evaluate possible cell and hemolysis contamination in samples. Were any other quality controls of the RNA performed? (ok)
ANS: We thank the reviewer for the comment. In order to deplete cell contamination and hemolysis, mouse sera were obtained after centrifuge at 5,000x rpm for 10 minutes, and hemolysis samples (color discoloration) are discarded. The RNA quality was measured by Spectrometry. The ratio of absorbance between DNA and protein (A260/280) is not lower than 1.8. The RNA integrity was determined using Bioanalyzer period to experimental set up. Most of our RNA are range between 25-50 bp.
Hence, we added it in the method section as follows
“Because the miR results could be interfered with by cell contamination and red blood cell rupture [86], sera were collected after centrifuged at 5,000 x rpm for 10 minutes before determination of hemolysis and RNA purity. Then, sample discoloration was excluded to reduce the interference from hemolysis. The ratio of absorbance at 260 nm and 280 nm is used to assess the purity of RNA and the RNA samples with an absorbance ratio (absorbance at 260/ absorbance at 280) higher than 2.0 were used.”
Reviewer 3
The manuscript entitled "Sepsis encephalopathy is partly mediated by miR370-3p induced mitochondrial injury but attenuated by BAM15 in cecal ligation and puncture sepsis mice" by Hiengrach et al. is a well-designed and well-performed study providing a convincing line of evidence that BAM15, a mitochondrial uncoupler, can improve the sepsis-related outcomes acting on brain-derived miR370-3p downregulation, and stimulating the anti-inflammatory macrophages polarization.
I have neither meritocracy nor methodologic concerns regarding the study. The manuscript should be, however, carefully re-read for typos and misspellings. Some abbreviations are explained in the main text two times, and some sentences are repeated. Thus, the manuscript should be unified. (ok)
ANS: We apologize for our mistake and correct it accordingly. We correct the miss-spellings, check abbreviation and the repeated sentences.
The result section should be rearranged. It is sometimes hard to follow the information in the text and to find the proper figure demonstrating the described data. It would be easier for a reader if the figures appear directly after the sentence in which they are cited. Maybe it would help if the results section is combined with the discussion. (ok)
ANS: We apologize for the unclear presentation and move the figure

Reviewer 2 Report
Title: Sepsis Encephalopathy is Partly Mediated by miR370-3p Induced Mitochondrial Injury but Attenuated by BAM15 in Cecal ligation and Puncture Sepsis Mice
Main point
Despite recognition of the importance of sex and gender in most areas of research, important knowledge gaps persist owing to the general orientation of scientific attention to one sex category. The gap in the representation of females in studies on human subjects, but also in animal models has been well documented.
Title and abstract. If only one sex is included in the study, the title and the abstract should specify the sex of animals. Authors should indicate if and why the study model was based on one sex. The article’s title should specify this fact by including “in males” in the title and abstract.
Methods. Authors should justify the reasons for any exclusion of female mice. Methodological choices about sex in relation to your study should be reported and justified in the same way as other methodological choices.
Discussion Authors should further discuss the implications of the lack of female mice analysis on the interpretation of the results.
Minor points
Materials and methods:
Page 17 Line 526: the sentence “The cDNA template and target primers based on the ΔΔCT method (2-∆∆Ct) relative to the β-actin housekeeping gene were conducted” is incorrect. Maybe you meant, “Relative expression was calculated based on the ΔΔCT method (2-∆∆Ct) relative to the β-actin housekeeping gene”.
Moreover, Figure 9 shows gene expression of what? 2-∆∆Ct respect to the control? However, controls are not =1 in the graphs.
For the analysis of miRNA in serum, the authors should describe details of the methods used to evaluate possible cell and hemolysis contamination in samples. Was any other quality controls of the RNA performed?
Author Response
Reviewer 2
Main point
Despite recognition of the importance of sex and gender in most areas of research, important knowledge gaps persist owing to the general orientation of scientific attention to one sex category. The gap in the representation of females in studies on human subjects, but also in animal models has been well documented. (ok)
ANS: We thank the reviewer for the comment. We usually use male mice in sepsis just because of our habituate protocol and there was a possible less severe sepsis in female mice. We will note this limitation in several parts of our manuscript.
Title and abstract. If only one sex is included in the study, the title and the abstract should specify the sex of the animals. Authors should indicate if and why the study model was based on one sex. The article’s title should specify this fact by including “in males” in the title and abstract. (ok)
ANS: We agree with the reviewer and put “male” in the title as follows “Sepsis encephalopathy is partly mediated by miR370-3p induced mitochondrial injury but attenuated by BAM15 in cecal ligation and puncture sepsis male mice”.
Methods. Authors should justify the reasons for any exclusion of female mice. Methodological choices about sex in relation to your study should be reported and justified in the same way as other methodological choices. (ok)
ANS: We thank the reviewer for the comment and put a remark in this aspect as follows “Only male mice were used in the experiments due to the well-known characterization of sepsis in male mice from our group, the effect of gender difference in sepsis should be considered for the clinical translation.”.
Discussion Authors should further discuss the implications of the lack of female mice analysis on the interpretation of the results. (ok)
ANS: We thank the reviewer for the comment and discuss this limitation as following “However, the conclusions are based on the sepsis model in the male gender which is possibly more severe than the female sepsis. The exploration of only male sepsis might limit the clinical translation from these results. Further studies are needed to characterize the role of miR-370-3p in sepsis conditions.”
Minor points
Materials and methods:
Page 17 Line 526: the sentence “The cDNA template and target primers based on the ΔΔCT method (2-∆∆Ct) relative to the β-actin housekeeping gene were conducted” is incorrect. Maybe you meant, “Relative expression was calculated based on the ΔΔCT method (2-∆∆Ct) relative to the β-actin housekeeping gene”. (ok)
ANS: We apologize for our mistake and correct it accordingly.
Moreover, Figure 9 shows gene expression of what? 2-∆∆Ct respect to the control? However, controls are not =1 in the graphs. (ok)
ANS: We apologize for our mistake and correct figure 9.
For the analysis of miRNA in serum, the authors should describe details of the methods used to evaluate possible cell and hemolysis contamination in samples. Were any other quality controls of the RNA performed? (ok)
ANS: We thank the reviewer for the comment. In order to deplete cell contamination and hemolysis, mouse sera were obtained after centrifuge at 5,000x rpm for 10 minutes, and hemolysis samples (color discoloration) are discarded. The RNA quality was measured by Spectrometry. The ratio of absorbance between DNA and protein (A260/280) is not lower than 1.8. The RNA integrity was determined using Bioanalyzer period to experimental set up. Most of our RNA are range between 25-50 bp.
Hence, we added it in the method section as follows
“Because the miR results could be interfered with by cell contamination and red blood cell rupture [86], sera were collected after centrifuged at 5,000 x rpm for 10 minutes before determination of hemolysis and RNA purity. Then, sample discoloration was excluded to reduce the interference from hemolysis. The ratio of absorbance at 260 nm and 280 nm is used to assess the purity of RNA and the RNA samples with an absorbance ratio (absorbance at 260/ absorbance at 280) higher than 2.0 were used.”
Reviewer 3
The manuscript entitled "Sepsis encephalopathy is partly mediated by miR370-3p induced mitochondrial injury but attenuated by BAM15 in cecal ligation and puncture sepsis mice" by Hiengrach et al. is a well-designed and well-performed study providing a convincing line of evidence that BAM15, a mitochondrial uncoupler, can improve the sepsis-related outcomes acting on brain-derived miR370-3p downregulation, and stimulating the anti-inflammatory macrophages polarization.
I have neither meritocracy nor methodologic concerns regarding the study. The manuscript should be, however, carefully re-read for typos and misspellings. Some abbreviations are explained in the main text two times, and some sentences are repeated. Thus, the manuscript should be unified. (ok)
ANS: We apologize for our mistake and correct it accordingly. We correct the miss-spellings, check abbreviation and the repeated sentences.
The result section should be rearranged. It is sometimes hard to follow the information in the text and to find the proper figure demonstrating the described data. It would be easier for a reader if the figures appear directly after the sentence in which they are cited. Maybe it would help if the results section is combined with the discussion. (ok)
ANS: We apologize for the unclear presentation and move the figure

Reviewer 3 Report
The manuscript entitled "Sepsis encephalopathy is partly mediated by miR370-3p induced mitochondrial injury but attenuated by BAM15 in cecal ligation and puncture sepsis mice" by Hiengrach et al. is a well-designed and well-performed study providing a convincing line of evidence that BAM15, a mitochondrial uncoupler, can improve the sepsis-related outcomes acting on brain-derived miR370-3p downregulation, and stimulating the anti-inflammatory macrophages polarization.
I have neigther meritoric nor methodologic concerns regarding the study. The manuscript should be, however, carefully re-read for typos and misspelings. Some abbreviations are explained in the main text two times, and some sentences are repeated. Thus, the manuscript should be unified.
The result section should be rearranged. It is sometimes hard to follow the information in the text and to find the proper figure demonstrating the described data. It would be easier for a reader if the figures appear directly after the sentence in which they are cited. Maybe it would help if the results section is combined with the discussion.
Author Response
Reviewer 2
Main point
Despite recognition of the importance of sex and gender in most areas of research, important knowledge gaps persist owing to the general orientation of scientific attention to one sex category. The gap in the representation of females in studies on human subjects, but also in animal models has been well documented. (ok)
ANS: We thank the reviewer for the comment. We usually use male mice in sepsis just because of our habituate protocol and there was a possible less severe sepsis in female mice. We will note this limitation in several parts of our manuscript.
Title and abstract. If only one sex is included in the study, the title and the abstract should specify the sex of the animals. Authors should indicate if and why the study model was based on one sex. The article’s title should specify this fact by including “in males” in the title and abstract. (ok)
ANS: We agree with the reviewer and put “male” in the title as follows “Sepsis encephalopathy is partly mediated by miR370-3p induced mitochondrial injury but attenuated by BAM15 in cecal ligation and puncture sepsis male mice”.
Methods. Authors should justify the reasons for any exclusion of female mice. Methodological choices about sex in relation to your study should be reported and justified in the same way as other methodological choices. (ok)
ANS: We thank the reviewer for the comment and put a remark in this aspect as follows “Only male mice were used in the experiments due to the well-known characterization of sepsis in male mice from our group, the effect of gender difference in sepsis should be considered for the clinical translation.”.
Discussion Authors should further discuss the implications of the lack of female mice analysis on the interpretation of the results. (ok)
ANS: We thank the reviewer for the comment and discuss this limitation as following “However, the conclusions are based on the sepsis model in the male gender which is possibly more severe than the female sepsis. The exploration of only male sepsis might limit the clinical translation from these results. Further studies are needed to characterize the role of miR-370-3p in sepsis conditions.”
Minor points
Materials and methods:
Page 17 Line 526: the sentence “The cDNA template and target primers based on the ΔΔCT method (2-∆∆Ct) relative to the β-actin housekeeping gene were conducted” is incorrect. Maybe you meant, “Relative expression was calculated based on the ΔΔCT method (2-∆∆Ct) relative to the β-actin housekeeping gene”. (ok)
ANS: We apologize for our mistake and correct it accordingly.
Moreover, Figure 9 shows gene expression of what? 2-∆∆Ct respect to the control? However, controls are not =1 in the graphs. (ok)
ANS: We apologize for our mistake and correct figure 9.
For the analysis of miRNA in serum, the authors should describe details of the methods used to evaluate possible cell and hemolysis contamination in samples. Were any other quality controls of the RNA performed? (ok)
ANS: We thank the reviewer for the comment. In order to deplete cell contamination and hemolysis, mouse sera were obtained after centrifuge at 5,000x rpm for 10 minutes, and hemolysis samples (color discoloration) are discarded. The RNA quality was measured by Spectrometry. The ratio of absorbance between DNA and protein (A260/280) is not lower than 1.8. The RNA integrity was determined using Bioanalyzer period to experimental set up. Most of our RNA are range between 25-50 bp.
Hence, we added it in the method section as follows
“Because the miR results could be interfered with by cell contamination and red blood cell rupture [86], sera were collected after centrifuged at 5,000 x rpm for 10 minutes before determination of hemolysis and RNA purity. Then, sample discoloration was excluded to reduce the interference from hemolysis. The ratio of absorbance at 260 nm and 280 nm is used to assess the purity of RNA and the RNA samples with an absorbance ratio (absorbance at 260/ absorbance at 280) higher than 2.0 were used.”
Reviewer 3
The manuscript entitled "Sepsis encephalopathy is partly mediated by miR370-3p induced mitochondrial injury but attenuated by BAM15 in cecal ligation and puncture sepsis mice" by Hiengrach et al. is a well-designed and well-performed study providing a convincing line of evidence that BAM15, a mitochondrial uncoupler, can improve the sepsis-related outcomes acting on brain-derived miR370-3p downregulation, and stimulating the anti-inflammatory macrophages polarization.
I have neither meritocracy nor methodologic concerns regarding the study. The manuscript should be, however, carefully re-read for typos and misspellings. Some abbreviations are explained in the main text two times, and some sentences are repeated. Thus, the manuscript should be unified. (ok)
ANS: We apologize for our mistake and correct it accordingly. We correct the miss-spellings, and check abbreviations and repeated sentences.
The result section should be rearranged. It is sometimes hard to follow the information in the text and to find the proper figure demonstrating the described data. It would be easier for a reader if the figures appear directly after the sentence in which they are cited. Maybe it would help if the results section is combined with the discussion. (ok)
ANS: We apologize for the unclear presentation and move the figure

Round 2
Reviewer 2 Report
The authors answered all my questions.The manuscript can be published in present form.